# RNAseq-Based Carboxylesterase *Nl-EST1* Gene Expression Plasticity Identification and Its Potential Involvement in Fenobucarb Resistance in the Brown Planthopper *Nilaparvata lugens*

**DOI:** 10.3390/insects15100743

**Published:** 2024-09-26

**Authors:** Murtaza Khan, Changhee Han, Nakjung Choi, Juil Kim

**Affiliations:** 1Agriculture and Life Science Research Institute, Kangwon National University, Chuncheon 24341, Republic of Korea; murtazakhan@kangwon.ac.kr; 2Interdisciplinary Graduate Program in Smart Agriculture, Kangwon National University, Chuncheon 24341, Republic of Korea; 201711031@kangwon.ac.kr; 3Crop Foundation Research Division, National Institute of Crop Science, Rural Development Administration, Wanju 55365, Republic of Korea; njchoi@korea.kr; 4Department of Plant Medicine, Division of Bio-resource Sciences, College of Agriculture and Life Science, Kangwon National University, Chuncheon 24341, Republic of Korea

**Keywords:** brown planthopper, carbamate resistance, RNA-seq analysis, *Nl-EST1*, metabolic resistance

## Abstract

**Simple Summary:**

Carbamate insecticides like fenobucarb have been used in the Republic of Korea since 1981 to control brown planthopper (BPH). However, their excessive use has resulted in increased resistance in BPH *Nilaparvata lugens*. Therefore, in our current study, we aimed to investigate the main cause of higher resistance to fenobucarb in BPH19 compared to BPH80 and BPH15. We examined mutations in the *acetylcholinesterase 1* gene, the target gene of carbamate, but did not find any previously reported mutations. However, through RNA-seq analysis, we discovered a strong correlation between the level of resistance development and the expression of the *N. lugens carboxylesterase 1*, *Nl-EST1* gene in BPH19 compared to BPH80 and BPH15. This suggests that *Nl-EST1* could be utilized as an expression marker for diagnosing resistance and integrated resistance management of *N. lugens*.

**Abstract:**

Carbamate insecticides have been used for over four decades to control brown planthopper, *Nilaparvata lugens*, but resistance has been reported in many countries, including the Republic of Korea. The bioassay results on resistance to fenobucarb showed that the LC_50_ values were 3.08 for the susceptible strain, 10.06 for the 2015 strain, and 73.98 mg/L for the 2019 strain. Compared to the susceptible strain, the 2015 and 2019 strains exhibited resistance levels 3.27 and 24.02 times higher, respectively. To elucidate the reason for the varying levels of resistance to fenobucarb in these strains, mutations in the *acetylcholinesterase 1* (*ACE1*) gene, the target gene of carbamate, were investigated, but no previously reported mutations were confirmed. Through RNA-seq analysis focusing on the expression of detoxification enzyme genes as an alternative resistance mechanism, it was found that the carboxylesterase gene *Nl-EST1* was overexpressed 2.4 times in the 2015 strain and 4.7 times in the 2019 strain compared to the susceptible strain. This indicates a strong correlation between the level of resistance development in each strain and the expression level of *Nl-EST1*. Previously, *Nl-EST1* was reported in an organophosphorus insecticide-resistant strain of Sri Lanka 2000. Thus, *Nl-EST1* is crucial for developing resistance to organophosphorus and carbamate insecticides. Resistance-related genes such as *Nl-EST1* could serve as expression markers for resistance diagnosis, and can apply to integrated resistance management of *N. lugens*.

## 1. Introduction

Rice is a widely consumed staple grain that is rich in fiber, energy, minerals, vitamins, and various biomolecules [1]. Research has shown that different parts of rice offer numerous health benefits in both preclinical and clinical studies. As a result, the constituents of rice are becoming increasingly popular for use in creating pharmaceutical adjuvants, food additives, and dietary supplements [2]. Furthermore, rice is a suitable host for a wide range of insects. Insect pests have always been a major issue for rice, causing significant yield loss and deterioration in grain quality. Among these pests, the brown planthopper (BPH, *Nilaparvata lugens* Stål, 1854) (Hemiptera: Delphacidae) is the most destructive, leading to about 20% to 80% yield loss and an annual economic loss of around USD 300 million in Asia [3]. BPH harms rice crops by laying eggs and extracting sap from the xylem and phloem tissues, resulting in “hopper burn”. In addition, BPH indirectly causes harm by transmitting viral diseases like grassy stunt virus and ragged stunt virus [4,5].

In the Republic of Korea, *N. lugens*, a significant migratory insect species originating from China, seriously threatens agriculture [6]. This pest is primarily controlled using organophosphate (OP), carbamate (CB), pyrethroid, nereistoxin, and neonicotinoid insecticides. OP and CB insecticides have been employed for 50 years to manage *N. lugens* populations. However, the extensive use of OP and CB insecticides has led *N. lugens* to develop resistance in Japan, Taiwan, Solomon Islands, Philippines, Malaysia, and the Republic of Korea [7].

Metabolic resistance in insects to insecticides involves the detoxification of chemicals through the overproduction of specific enzymes that break down the insecticides before they can reach and bind to their target sites. This mechanism relies heavily on enzymes such as monooxygenases (cytochrome P450 monooxygenases, CYP), hydrolases (such as esterases, ESTs), and transferases (glutathione-S-transferase, GST). In the first phase of metabolism, monoxygenases and hydrolases modify xenobiotics, making them more reactive. In the second phase, GSTs conjugate endogenous molecules such as glutathione to these metabolites, increasing their polarity and hydrophilicity. This process facilitates their excretion and aids in detoxification [8]. Esterases, such as E4 and FE4, degrade ester bonds in insecticides like OP and CB, often due to gene amplification or upregulation [9]. Overexpression of CYPs, particularly in species like *Myzus persicae,* is linked to neonicotinoid resistance through increased gene copy numbers and mutations [10]. Similarly, GSTs are pivotal in detoxifying endogenous and exogenous compounds, with enhanced expression linked to resistance to various pests [11]. Mutation-based resistance occurs when the target insecticide site is modified, reducing binding efficiency. Notable examples include nicotinic acetylcholine receptors, where single point mutations in the D-loop region or alterations in subunits like Mdα2 and Mdα6 in houseflies confer resistance to neonicotinoids and spinosad [12,13,14]. Additionally, modified acetylcholine esterase due to gene mutations results in insensitivity to OP and CB, with specific mutations like G262V in *Musca domestica* showing strong resistance [15]. Both metabolic and mutation-based resistance exemplify pests’ adaptive strategies to overcome chemical control measures, necessitating the ongoing research and development of novel insecticides.

Carboxyl/choline esterases (CCEs), particularly carboxylesterases, constitute a diverse and widespread group of enzymes involved in a range of metabolic processes, including hormone metabolism, pheromone breakdown, detoxification of foreign substances, and hydrolysis of carboxyl esters in insecticides [16]. The enhanced ability of CCEs to detoxify is linked to resistance to various insecticides like organophosphates, carbamates, and pyrethroids [17]. Previous research indicates that heightened expression and activity of specific CCE are associated with increased resistance to insecticides across several insect species, such as *N. lugens* [18] and *M. persicae* [19,20].

Our research group identified the I4790M mutation in the *Spodoptera exigua* ryanodine receptor gene, noting that its correlation with resistance to diamide insecticides varied among the field-collected population in the Republic of Korea [21]. The authors proposed that transcription regulation could contribute to enhanced resistance in *S. exigua* against diamides in addition to this mutation. Similarly, Adelman, et al. [22] identified that the esterase-encoding genes *CE3959* and *CE21331* were significantly overexpressed in the highly resistant Richmond strain of *Cimex lectularius*, suggesting their role in esterase-mediated resistance. Zhu, et al. [23] also confirmed the overexpression of *CLCE21331* in resistance strains of *C. lectularius*, showing more than 50-fold upregulation in most field populations, indicating its critical role in pyrethroid resistance. In contrast, mutation in coding gene sequences is rarely reported globally and has not yet been observed in bed bugs, indicating its limited documentation and potential relevance [24]. Building on this, our current study aimed to investigate resistance to the carbamate insecticide fenobucarb in *N. lugens* in the Republic of Korea, focusing on transcriptomic changes.

## 2. Materials and Methods

### 2.1. Insects

The insecticide-susceptible strain (BPH80) of brown planthopper (BPH, *Nilaparvata lugens*) was shared by the National Academy of Agriculture Science, Rural Development Administration (RDA), Republic of Korea. This strain was collected from the field in 1980 and grown in the laboratory without insecticide exposure for over 40 years. In addition to the insecticide-susceptible strain collected in 1980, they were collected from Wando (34°22′06″ N, 126°43′27″ E) and Goseong (34°57′19″ N, 128°20′38″ E) in 2015 (BPH15) and 2019 (BPH19), respectively, and reared in the National Institute of Crop Science, RDA. All bioassays and RNA extractions were performed after 2019 when BPH19 was secured, and there was no more selection by insecticide treatment in the lab. BPH80 was maintained for about 40 years, and BPH15 for about 5 years without insecticide exposure in the lab. All experiments were performed on BPH19 within three generations after collection. Since all BPHs originated from field populations, we cannot know the extent and dose of pesticide exposure before field collection. However, after collection in the lab, BPHs were reared without pesticide exposure after morphological and molecular species identification using the mtCO1 marker. All BPHs were maintained under controlled conditions at a temperature of 25 ± 1 °C, relative humidity of 60 ± 5%, and a photoperiod regime of 14 h of light and 10 h of darkness, as previously reported [7].

### 2.2. Bioassay for Fenobucarb Resistance

Bioassays were performed based on the IRAC (Insecticide Resistance Action Committee) susceptibility test method 005 with some modifications (www.irac-online.org accessed on 20 June 2023) for more than 30 wingless female BPH adults. Fenobucarb (BPMC, emulsifiable concentrate formulation of 50% *w*/*v*) was used for bioassay. After diluting fenobucarb to various concentrations, rice seedlings of ten days were dipped into the diluted solution for 10 s, and then ten wingless female BPH adults per treatment were transferred. All experiments were conducted on more than three biological replications (*n* > 30 per concentration). More detailed bioassay methods were followed by IRAC susceptibility test method 005.

Using the SAS program based on the Probit model (SAS Institute 9.1, Cary, NC, USA), concentration-based mortality after two days of fenobucarb exposure was estimated to determine the median lethal concentration (LC_50_) and 95% confidence limits (CLs). RR (resistance ratio) was computed by dividing the LC_50_ value of the tested field strain by that of the susceptible strain, BPH80.

### 2.3. RNA and DNA Extraction

Total RNAs were extracted from the adults of each strain of *N. lugens* within 12 h after emergence, with each sample containing twenty wingless female adults as a biological replicate. RNA extraction was conducted using the RNeasy Mini Kit (Qiagen, Hilden, Germany), according to the manufacturer’s instructions. The RNA was validated and quantified using an Agilent 2200 TapeStation (Agilent Technologies, Santa Clara, CA, USA), and RNA integrity was confirmed by running samples on a 1% agarose gel using electrophoresis. For the reverse transcription reaction, we utilized the SuperiorScript III cDNA Synthesis Kit (Enzynomics, Daejeon, Republic of Korea). The total RNAs and synthesized cDNA were stored at −70 °C before the next experiments. Furthermore, genomic DNA (gDNA) was extracted from the twenty wingless female adults of each *N. lugens* within 12 h after emergence, with each sample containing twenty adults as a biological replicate using DNeasy Blood & Tissue (Qiagen) following the manufacturer’s instructions and quantified using Nanodrop (Nanodrop Technologies, Wilmington, DE, USA).

### 2.4. Mutation Survey

Following specific thermal conditions, the cDNA and gDNA underwent PCR to survey the mutations in *ACE1* and *Nl-EST1*, using the ProFlex PCR System (ThermoFisher Scientific Inc., Waltham, MA, USA) with KOD FX polymerase (Toyobo Life Science, Osaka, Japan) with the appropriate primer combinations and PCR conditions. Used primer sets are listed in Table 1. The PCR products were directly sequenced (Macrogen, Seoul, Republic of Korea) and the chromatograms were analyzed for mutations, following the methodology described earlier [21]. Pooled gDNA (extracted from 20 wingless female adults) rather than individual level was used to confirm the four previously reported mutation sites. However, cDNA was used to verify the presence or absence of mutations within the entire ORF.

### 2.5. RNA-Seq Analysis

Total RNA was extracted from the adults of all strains, namely, BPH80, BPH15, and BPH19, of *N. lugens*, within 12 h after emergence, with each sample containing twenty wingless female adults as a biological replicate. RNA extraction was conducted using the RNeasy Mini Kit (Qiagen, Hilden, Germany) according to the manufacturer’s instructions. The RNA was validated and quantified using an Agilent 2200 TapeStation (Agilent Technologies, Santa Clara, CA, USA), and RNA integrity was confirmed by running samples on a 1% agarose gel using electrophoresis. RNA-seq libraries were prepared using the TruSeq RNA sample Prep Kit v2 (Illumina, San Diego, CA, USA). Samples were sequenced on the Hiseq4000 plDEGatform using TruSeq 3000/4000 SBS Kit v3 (Macrogen, Seoul, Republic of Korea).

The nine RNA-seq raw sequences were initially processed using Trimmomatic v0.38 to remove low-quality sequences (Q30) and adapters from the raw data [25]. The quality of the resulting trimmed reads was confirmed to be high using FastQC v0.11.7 (http://www.bioinformatics.babraham.ac.uk/projects/fastqc accessed on 1 November 2023).

### 2.6. Clean Read Assembly and Unigene Construction

Trimmed reads from all samples were merged into each assembly group to construct transcriptome references. The merged data were assembled using the Trinity version (r20140717) program, utilized for de novo transcriptome assembly [26]. This process results in transcript fragments called contigs. The longest contigs were clustered into nonredundant transcripts, referred to as unigenes, using the CD-HIT-EST program provided by CD-HIT v4.6 [27].

### 2.7. Functional Annotation

Protein sequences were generated by predicting ORFs from unigenes using TransDecoder v3.0.1, and the generated unigenes and protein sequences were used for annotation and expression analysis. To perform functional annotation, we used BLASTN from NCBI BLAST v2.9.0+ and BLASTX of DIAMOND v0.9.21 software with an E-value default cutoff of 1.0 × 10^−5^ [28], against the Kyoto Encyclopedia of Genes and Genomes (KEGG), NCBI Nucleotide (NT), Pfam, Gene Ontology (GO), NCBI nonredundant Protein (NR), UniProt, and EggNOG [29].

### 2.8. Differential Gene Expression Analysis

Trimmed reads for each sample were aligned to the assembled unigene as reference using Bowtie v1.1.2. For the differentially expressed gene analysis, the abundances of unigenes across samples were estimated into the read count as an expression measure by the RSEM v1.3.1 algorithm [30]. For nine samples, if more than one read count value was 0, it was not included in the analysis. To reduce systematic bias, we estimated the size factors from the count data and applied relative log expression (RLE) normalization with DESeq2 v1.28.1 (https://www.bioconductor.org/packages/release/bioc/html/DESeq2.html accessed on 1 November 2023). Differentially expressed genes (DEGs) analysis and statistical analysis were performed using log2foldchange (FC) and nbinomWaldTest per comparison pair using DESeq2 (|FC| > 2 and nbinomWaldTest raw *p*-value < 0.05).

### 2.9. Orthologous Cluster Analysis

Using the web program Orthovenn3, orthologous and particular genes among the predicted protein unigenes of all BPH were investigated and displayed [31]. Venn diagrams were used for the comparative analysis and visualization of orthologous gene clusters and distinct genes that are particular to each BPH. Then, protein sequences classified into strain-specific clusters were subjected to GO analysis to evaluate strain-specific biological roles.

## 3. Results

### 3.1. Bioassay

The rice seedling dip method-based bioassays demonstrated significant variations in susceptibility to the carbamate insecticide fenobucarb among the susceptible strain (BPH80) and field strains (BPH15 and BPH19). The LC_50_ values were 3.08 for the BPH80 strain, 10.06 for the BPH15 strain, and 73.98 for the BPH19 strain. Compared to BPH80, BPH15 and BPH19 exhibited resistance levels 3.27 times and 24.02 times higher, respectively. This indicates a progressive increase in resistance, with the order of susceptibility being BPH80 > BPH15 > BPH19.

These results highlight the growing resistance to fenobucarb and suggest that the BPH80 strain is highly susceptible, while BPH15 and BPH19 strains have developed significant resistance. The bioassay results are shown in Table 2.

### 3.2. Mutation Analysis

The comparison of amino acid sequence in the mutation-reported regions of the *ACE1* gene across four pest species, including *N. lugens*, *Laodelphax striatella*, *Sogatella furcifera* (Delphacidae), and *Nephotettix cincticeps* (Cicadellidae), showed a highly conserved region with minimal variation. The *ACE2* gene in *N. lugens* was used as an outgroup for the analysis (Figure 1A). As a result, no mutations were found in the known resistance-associated sites within the field strains (Figure 1B,C). These findings suggest that the fenobucarb resistance in these strains is not due to point mutations in the *ACE1* gene but may be related to transcriptomic changes of other genes. We checked for mutations in the *Nl-EST1*, but no nonsynonymous substitutions were found in BPH15 and BPH19 (Appendix A).

### 3.3. Raw and Trimmed Data Statistics of RNA-seq

To ensure the reliability and quality of sequencing data for genetic analysis, we performed sequencing on three distinct samples: BPH80, BPH15, and BPH19, with three biological replicates each. The quality of raw sequencing data was evaluated by determining the total number of bases, total reads, GC contents (%), and Q30 (%) for nine samples. The total read bases per sample ranged from 5.39 giga base pair (Gb) to 6.46 Gb, and total reads per sample ranged from 53.38 million to 64.03 million. The GC (%) content per sample ranged from 31 to 40. Additionally, Q30 (%) values were 95 and 96, as illustrated in Appendix A. The data indicate high-quality sequencing across all samples, with consistently high Q30 values, ensuring reliable downstream analysis.

Subsequently, the raw reads were processed using the Trimmomatic program to remove adapter sequences and low-quality bases. The total read bases per sample ranged from 5.26 Gb to 6.33 Gb, and the total reads per sample ranged from 52.42 million to 63.10 million (Appendix A). The GC (%) content per sample ranged from 31 to 40. Similarly, Q30 (%) values were 96, as illustrated in Appendix A. These high-quality trimmed data ensure accuracy and reliability for subsequent analysis.

### 3.4. De Novo Assembly of Unigene Sets

To accurately represent gene expression patterns and reduce any potential errors, we carried out a de novo assembly analysis on samples from BPH80, BPH15, and BPH19 following preprocessing. The purpose of this analysis was to create comprehensive transcriptomes for each sample. Detailed statistics for the initial assembled contig and unigene contig are provided in Table 3 and Appendix A, presenting data such as the number of genes, number of transcripts, %GC content, N50, average contig length, and total assembled bases (bp).

For the initial assembly, the emerged dataset revealed the highest number of genes at 148,234. In comparison, BPH19 had 111,118 genes, BPH80 had 92,362, and BPH15 had 79,691. The merged dataset also showed the highest number of transcripts at 191,287, surpassing BPH19 with 139,372, BPH80 with 115,494, and BPH15 with 97,626. The %GC content was highest in both the merged dataset and BPH19, both at 41%, while BPH80 and BPH15 had 39% and 38%, respectively. The N50 values mirrored this trend, with the merged assembly achieving an N50 value of 912, followed by BPH19 at 893, BPH80 at 810, and BPH15 with a lower value. The average contig length (bp) was greatest in the merged dataset at 626 bp, with BPH19 at 618 bp, BPH80 at 584 bp, and BPH15 at 577 bp. The total assembled bases were also highest in the merged dataset with 119,808,296 bp compared to 86,124,462 bp for BPH19, 67,502,190 bp for BPH80, and 56,350,010 bp for BPH15 (Appendix A).

In the unigene contig assembly, the emerged dataset contained 119,664 genes, which was more than BPH19 with 93,427 genes, BPH80 with 75,069, and BPH15 with 69,319. The number of transcripts followed a similar pattern, with the merged dataset having 119,664 transcripts, BPH19 with 93,427, BPH80 with 75,069, and BPH15 with 69,319. The %GC content remained highest in both the merged dataset and BPH19 at 41%, with BPH80 at 39% and BPH15 at 38%. The N50 values for unigene contig were also highest in the merged assembly at 896, with BPH19 at 860, BPH80 at 816, and BPH15 with the lowest value. The average contig length (bp) was greatest in the merged assembly at 611 bp, followed by BPH19 at 600 bp, BPH80 at 579 bp, and BPH15 at 560 bp. Regarding total assembled bases, the merged dataset had 73,153,339 bp, while BPH19 had 56,127,492 bp, BPH80 had 43,504,047 bp, and BPH15 had 38,844,156 bp (Table 3).

### 3.5. ORF Prediction

The analysis conducted on unigenes assembled from various datasets, including merge, BPH80, BPH15, and BPH19, revealed notable variations in the proportion of unigenes containing predicted ORFs across these datasets. In the merged dataset, comparing 119,664 total unigenes, 25.73% (30,788 unigenes) were found to have predicted ORFs. It is worth mentioning that the majority of these, 97.65% (30,064 unigenes), were predicted to have a single ORF, while only 2.35% (724 unigenes) had multiple ORFs (Table 4 and Appendix A).

In the BPH80 dataset, which consists of 75,069 total unigenes, 24.77% (18,593 unigenes) were identified as having predicted ORFs. Among these, 98.69% (18,349 unigenes) contained a single ORF, whereas a smaller fraction, 1.31% (244 unigenes), exhibited multiple ORFs. Similarly, the BPH15 dataset, with 69,319 total unigenes, showed that 22.91% (15,880 unigenes) were found to have predicted ORFs, of which 99.0% (15,721 unigenes) contained a single ORF, and only 1.0% (159 unigenes) had multiple ORFs. The BPH19 dataset, comprising 93,427 unigenes, demonstrated the highest percentage of predicted ORFs at 27.99% (26,147 unigenes). Among these, 98.7% (25,721 unigenes) had a single ORF, while 1.63% (426 unigenes) contained multiple ORFs (Table 4).

### 3.6. Transcriptomic DEG Analysis

RNA-seq data from all samples were analyzed based on the merged reference unigene set. The mapping efficiency of reads to the reference unigene set (merged) was evaluated for various samples, providing insight into the alignment quality and comprehensiveness of the unigene assembly (Appendix A). These results reflect the effectiveness of the unigene assembly in capturing the majority of the sequence reads, although a significant proportion of reads in some samples remained unmapped (Appendix A).

The results of our hierarchical clustering unveiled varying susceptibility among the field strains (BPH15 and BPH19) and BPH80 strain. BPH80 strain displayed higher susceptibility than BPH15 and BPH19 strains, which is illustrated in Figure 2A. Moreover, a principal component analysis (PCA) plot was created based on read counts from aligning each strain reads to the reference unigene. The PCA analysis indicates distinct gene expression patterns for all BPHs (Figure 2B). Similarly, volcano plots revealed a significant increase in gene expression levels in both BPH15 and BPH19 compared to BPH80, as shown in Figure 2C,D. Our current study evaluated differential gene expression in BPH80, BPH15, and BPH19 strains, indicating that these genes could a key role in the resistance of BPH19 to fenobucarb. However, further research is required to investigate whether these genes play a significant role in fenobucarb resistance. Our findings just suggest an association between gene expression patterns and resistance but do not establish causation.

### 3.7. Orthologous Cluster 

To gain a deeper understanding of the factors behind fenobucarb resistance, we analyzed the cluster count, protein count, singletons, and the number of specific or shared elements among all BPH. This analysis aimed to uncover the variations in gene expression and genetic diversity that may influence different levels of resistance. We noted significant disparities in cluster counts according to the level of fenobucarb resistance. The BPH19 strain, which exhibited higher resistance, showed a notably higher cluster count (16,708) compared to moderately resistant BPH15 (12,593) and the susceptible BPH80 (15,059) strains, as illustrated in Figure 3A. Similarly, BPH19 demonstrated a higher number of clusters, proteins, and singletons compared to BPH15 and BPH80. Specifically, BPH19 had 16,708 clusters, 26,591 proteins, and 9141 singletons. In contrast, BPH15 had 12,593 clusters, 16,045 proteins, and 3030 singletons, while BPH80 had 15,059 clusters, 18,842 proteins, and 3398 singletons, as shown in Figure 3B. Furthermore, we examined the presence of specific and shared elements among all BPHs. BPH19 displayed 454 specific elements, significantly more than BPH15 (21 specific elements) and BPH80 (20 specific elements). The analysis of shared elements revealed that BPH19 and BPH15 shared a total of 11,645 elements (2142 + 9503), BPH19 and BPH80 shared a total of 14,112 elements (9503 + 4609), and BPH15 and BPH80 shared 10,430 elements (9503 + 927), as depicted in Figure 3C. By comparing genetic homology based on orthologous genes, BPH19, which had the highest resistance, was more similar to BPH80 than BPH15 (Figure 3D,E). Additional GO analysis was performed on 20 genes of BPH80, 21 genes of BPH15, and 454 genes of BPH19 expressed in a strain-specific manner in Figure 3C. These genes were categorized into three gene ontology (GO) categories: biological process, cellular component, and molecular function, as shown in Appendix A. Our analysis reveals that, in comparison to BPH80 and BPH15, BPH19 exhibited a higher representation across all three GO categories. Specifically, BPH19 demonstrated an enhanced number of unigenes associated with biological processes, cellular components and molecular functions, indicating a broader functional diversity in this genotype. The correlation matrix data revealed a strong correlation among all replicates of each BPH, which further confirms the accuracy and reliability of our results (Appendix A). This consistency across replicates underscores the validity of our findings related to gene expression and insecticide resistance.

### 3.8. DEG Analysis of Detoxification Genes

In addition to the broad transcriptomic analysis (Section 3.6), we conducted a focused differential gene expression (DGE) analysis specifically targeting five detoxification gene families such as CYP, CCE, GST, uridine 5′-diphosphate-glucosyltransferase (UGT), ATP-binding cassette transporter (ABC), and cuticular protein (CP). This analysis allowed us to dissect the role of detoxification pathways in fenobucarb resistance more precisely. No correlation was found between the resistance level of BPH19 for the 34 out of 54 total CYP family genes, as well as all GSTs, UGTs, and ABCs genes analyzed (Figure 4). Conversely, a strong correlation was observed between resistance levels and the expression of CCEs in PBH15, especially in BPH19, compared to BPH80 (Figure 4B).

To further confirm the correlation between the expression of CCEs and the resistance levels, we calculated fragment per kilobase of transcript per million map reads (FPKM) values using the number of RNA-seq reads mapped to gene sequences for gene expression profiling. We examined the FPKM values to assess the expression of several genes from the *CCE6* subfamily, *E4* Type *CCE*, and *CCE2* subfamily. The primary objective of this analysis was to identify the key gene or genes involved in enhancing resistance in *N. lugens* against fenobucarb (Figure 5). Our results revealed a highly significant increase in the expression of *E4* type *CCE* gene *Nl-EST1* in BPH19, followed by BPH15, compared to BPH80 after fenobucarb exposure, as illustrated in Figure 5A. Conversely, no significant changes were observed in the expression of the *CCE6* and *CCE2* subfamily genes among all BPHs (Figure 5A). These findings indicate a strong correlation between the expression of the E4 type CCE gene *Nl-EST1* and heightened resistance levels. Resistance in BPHs has significantly increased, with BPH19 exhibiting the highest resistance to fenobucarb, followed by BPH15, while BPH80 remains highly sensitive to fenobucarb resistance.

## 4. Discussion

Rice, as a globally vital staple food, faces significant threats from insects such as *N. lugens*, especially in Asia. This pest is highly destructive to rice plants as it feeds on their sap, leading to significant damage and the transmission of viral diseases, resulting in reduced crop yields. Plants have developed several defense mechanisms, but chemical control remains a key method to mitigate *N. lugens* infestations [32,33]. However, repeated and improper use of insecticides, including carbamates like fenobucarb, has contributed to a notable increase in insect resistance, posing a significant global challenge for formers and scientists [34].

In the current study, we investigated fenobucarb resistance in *N. lugens* strains from the Republic of Korea collected in 1980, 2015, and 2019. Our bioassay results revealed a gradual increase in the resistance of *N. lugens* to fenobucarb, with the BPH80 strain being the most susceptible, while the BPH19 strain displayed the highest resistance, showing a 24.2-fold increase compared to the BPH80 strain. These results align with prior studies that have noted an escalation in *N. lugens* resistance to carbamate [7,35,36]. Our current results, along with previous findings, suggest that insects develop increased resistance to insecticides over time. However, while our bioassay data provided clear evidence of elevated resistance in BPH19 strain, a closer examination of the underlying mechanisms is crucial to fully understand this phenomenon.

Previous studies have identified two main mechanisms of insecticide resistance: target-site resistance and metabolic resistance. Target-site resistance is frequently linked with mutations in genes such as *ACE1* and has been observed in *N. lugens* and other insects following exposure to carbamates [35]. Mutations such as F331H and I332L have been linked to increased resistance to carbamates in other *N. lugens* strains. These mutations change the *ACE1* active site, reducing its sensitivity to inhibition by carbamates and, thus, promoting resistance. However, in the current study, sequence analysis of the *ACE1* gene in the Republic of Korean *N. lugens* strains did not reveal any mutations in the conserved regions. This suggests that fenobucarb resistance in these strains is not driven by typical target-site mutations (Figure 1). Our findings suggest that another resistance mechanism is also involved in addition to target-site resistance in BPH19 strains against fenobucarb. Our supposition is supported by previous research, which reported that metabolic resistance is also recognized as a key mechanism contributing to the development of insecticide resistance in insects [21].

The absence of *ACE1* mutations in all strains shifted our attention towards metabolic resistance in the Republic of Korean *N. lugens* against fenobucarb. Metabolic resistance is accelerated by various detoxification enzymes, including CYPs, ESTs, GSTs, UGTs, ABCs, and CPs, which play pivotal roles in insecticide resistance [37,38]. Our RNA-seq analysis also showed a significant increase in the upregulation of genes from the EST and CP families in the BPH19 strain compared to BPH80 and BPH15 strains, suggesting that elevated metabolic activity of EST and CP families might be involved in the heightened resistance in the BPH19 strain. Our current findings align with previous findings in which they reported that metabolic resistance is associated with the overexpression of detoxification genes [39].

Among the detoxification enzymes, carboxylesterases have been particularly implicated in developing resistance to insecticides such as carbamate. For instance, E4 and FE4 esterases have been reported to elevate the resistance against carbamate in other insect species, such as *M. persicae*, by hydrolyzing ester bonds in carbamate molecules, rendering them less toxic [40]. Our current findings also evaluated a strong correlation between the expression of the *Nl-EST1* gene and fenobucarb resistance in the BPH19 strain. This increased expression of *Nl-EST1* in the resistant BPH19 strain indicates that carboxylesterase-mediated detoxification plays a significant role in fenobucarb resistance, particularly involving carboxylesterase, which is a key factor in insecticide resistance development in this pest species. The results from Figure 2, Figure 4, and Figure 5 highlight the importance of using both global and targeted gene expression analysis to gain a better understanding of resistance mechanisms. In Figure 2, the global analysis of DEG shows distinct expression patterns among all BPH strains. BPH19 exhibited significant differences in gene expression compared to BPH80 and BPH15, suggesting its potential involvement in fenobucarb resistance. However, this broad analysis did not identify specific-resistance-related gene or pathways. On the other hand, the focused detoxification gene analysis (Figure 4 and Figure 5) showed a strong correlation between the upregulation of the E4-type CCE gene *Nl-EST1* and fenobucarb resistance, particularly in BPH19, followed by BPH15. This comparison demonstrates how a combination of global transcriptomic and targeted analysis can provide both a comprehensive overview of gene expression and a more precise identification of key resistance-related genes, such as *Nl-EST1*, in BPH strains.

Furthermore, genetic diversity analysis using the Orthovenn3 program revealed that the BPH19 strain showed more gene clusters and singletons than the BPH80 and BPH15 strains, reflecting greater genetic diversity. This genetic diversity among all BPH strains may provide a broader range of adaptive responses to insecticide exposure, facilitating the transcript accumulation of crucial resistance genes like *Nl-EST1*. The role of genetic variation in insecticide resistance has been also reported in previous studies, which have indicated that higher genetic diversity enables pest strains to evolve and adapt to selective pressures more effectively [41].

Conclusively, our findings suggest that fenobucarb resistance in *N. lugens* strains from the Republic of Korea is driven primarily by metabolic resistance mechanisms, particularly the upregulation of carboxylesterases like *Nl-EST1*. While target-site mutations in *ACE1* have been implicated in carbamate resistance in other strains, our study did not find evidence of such mutations, indicating that resistance in these strains may involve different pathways. These findings underscore the complexity of insecticide resistance and the need for further research to explore the broader spectrum of metabolic and nongenetic factors contributing to resistance in pest strains. Understanding these mechanisms is essential for developing more effective strategies to manage resistance and ensure sustainable pest management.

## 5. Conclusions

The use of carbamate insecticides such as fenobucarb to manage the rice pest BPH has been ongoing since their registration in the Republic of Korea in 1981. Unfortunately, the improper and excessive application of these insecticides has resulted in heightened resistance in *N. lugens* populations, presenting a considerable challenge for farmers and researchers seeking to comprehend and address this resistance. Our bioassay analysis indicated that BPH19 exhibited significantly greater resistance than BPH80 and BPH15. To delve into the reasons behind BPH19’s enhanced resistance to fenobucarb, we conducted an RNA-seq analysis. The transcriptomic analysis revealed that the upregulation of *E4* type esterase gene *Nl-EST1* contributed to the elevated resistance of BPH19 in comparison to BPH80 and BPH15. These findings suggest that *Nl-EST1* plays a pivotal role in the increased resistance of *N. lugens* to carbamate insecticides fenobucarb. Though our current investigation provides valuable insights into transcriptomic changes associated with fenobucarb resistance in *N. lugens* in the Republic of Korea, further validation through RT-PCR and synergist bioassays is needed to confirm the role of the *E4* types esterase gene *Nl-EST1* in contributing to the elevated resistance observed in the BPH19 strain.

## Figures and Tables

**Figure 1 insects-15-00743-f001:**
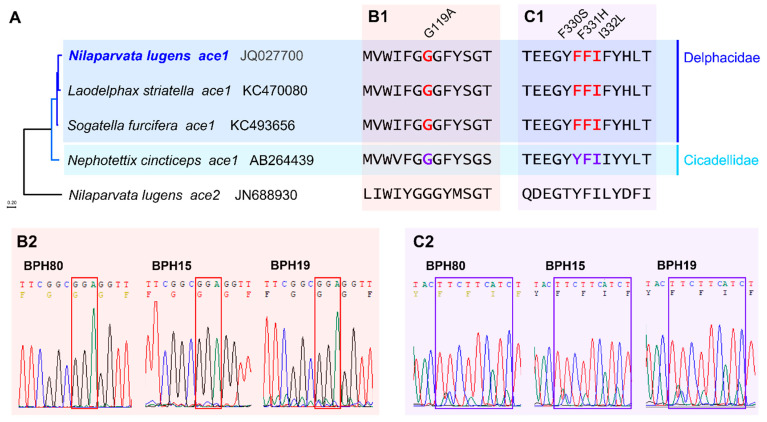
Mutation survey analysis of the *ACE1*, a well-known mechanism of resistance to carbamate insecticides in *Nilaparvata lugens*. (**A**) Phylogenetic relationship of the *ACE1* in *N. lugens*, *Laodelphax striatella*, *Sogatella furcifera* (Delphacidae), and *Nephotettix cincticeps* (Cicadellidae). The paralogous gene *ACE2* was used as an outgroup. (**B**) The aligned amino acid sequence of G119A mutation region, B1, and sequencing results of BPH80, 15, and 19, B2. (**C**) The aligned amino acid sequence of F330S, F331H, and I332L mutations and sequencing results.

**Figure 2 insects-15-00743-f002:**
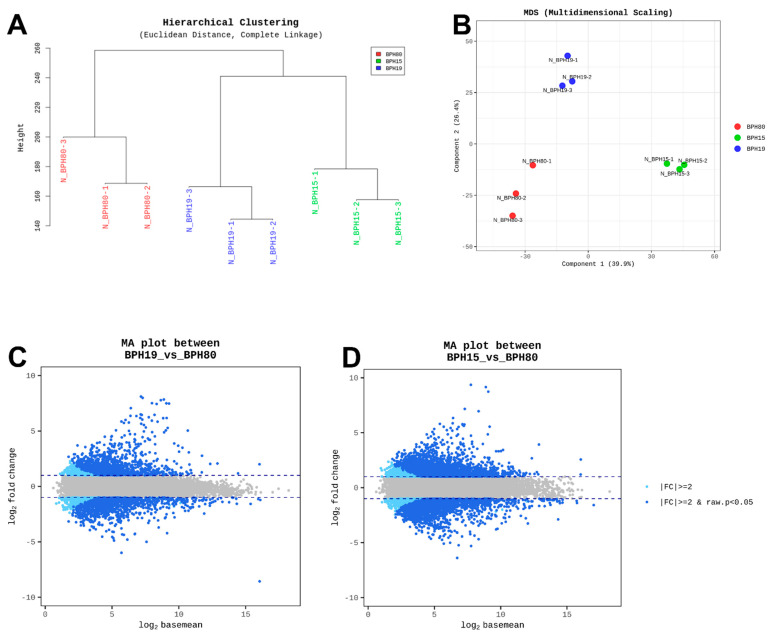
Differential gene expression (DGE) analysis results. (**A**) hierarchical clustering using Euclidean distance and complete linkage method across different *N. lugens* strains. (**B**) Multidimensional scaling plot displaying the genetic relationships between BPH strains. (**C**,**D**) Volcano plots comparing gene expression between BPH80, BPH15, and BPH19 strains.

**Figure 3 insects-15-00743-f003:**
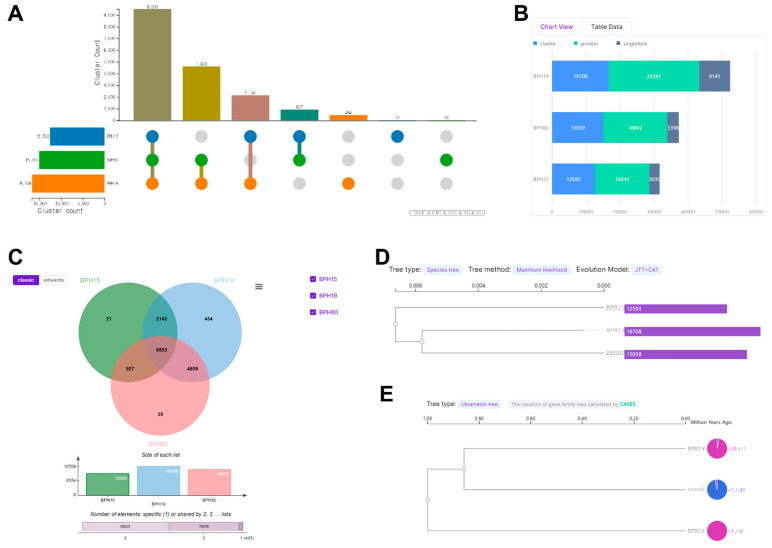
Orthovenn3 program confirming carrying resistance levels in *N. lugens* strains. (**A**) Cluster count A1 represents the number of genes expressed in all three strains. A2–4 represents the genes commonly expressed in two strains (BPH80 and BPH19; A2, BPH15 and BPH19; A3, BPH15 and BPH80; A4). A5–7 represent the number of genes expressed specifically in a strain (BPH19; A5, BPH15; A6, and BPH80; A7). (**B**) Number of clusters (blue color), proteins (green color), and singletons (ash color) in each strain. Singleton protein sequences did not cluster with other sequences and are shown as percentages of the total number of protein sequences in the strains. (**C**) The distribution of orthologous protein clusters of predicted protein sequences shown in different sections of the Venn diagram indicates the number of protein clusters in each species. (**D**) A phylogenetic-analysis-based species (strain) tree was generated using maximum likelihood with the JTT + CAT evolution model. (**E**) For the ultrametric tree, the variation of the gene family was calculated by CAFE5.

**Figure 4 insects-15-00743-f004:**
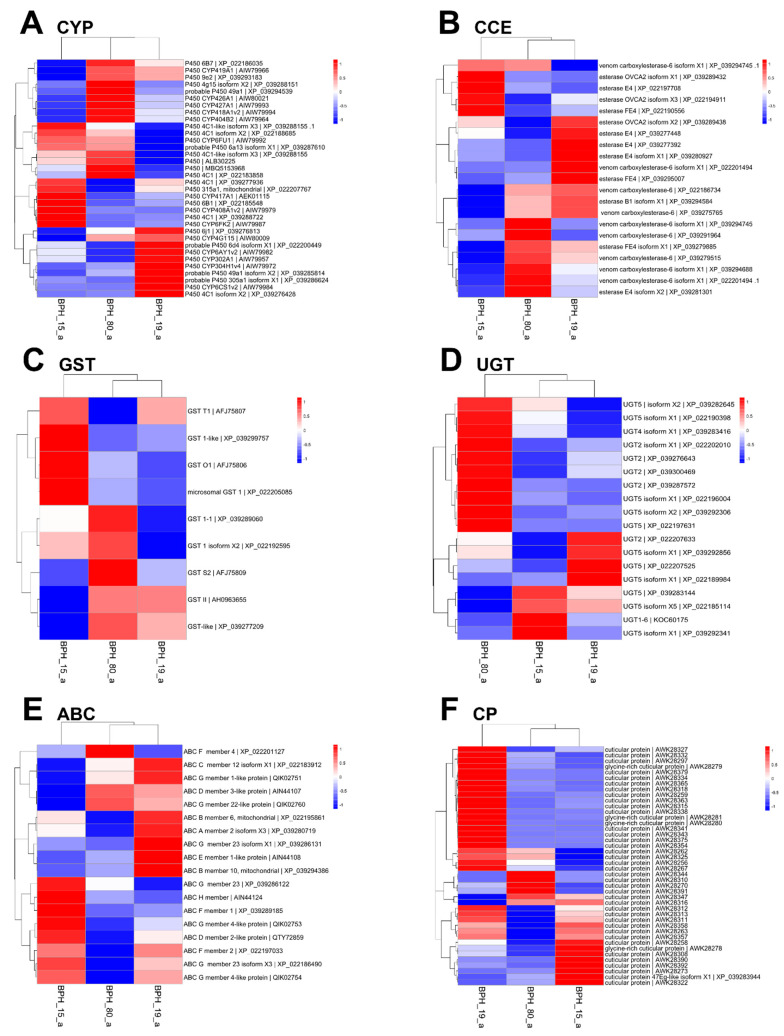
Heat map showing the expression of six major detoxification enzyme genes. (**A**–**F**) Cytochrome P450 (CYP), carboxyl/cholinesterase (CCE), glutathione S-transferase (GST), 5′-diphosphate-glucosyltransferase (UGT), ATP-binding cassette transporter (ABC), and cuticular protein (CP), respectively.

**Figure 5 insects-15-00743-f005:**
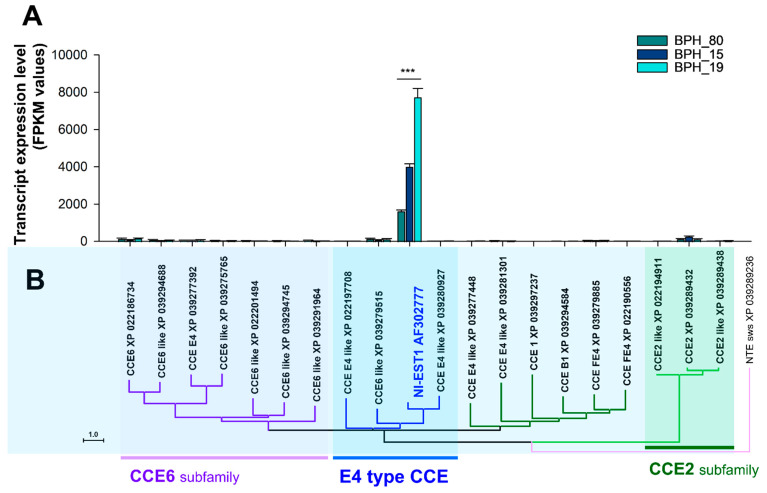
Expression comparison and phylogenetic analysis of the top 20 *CCE* genes with high expression levels. (**A**) The expression levels were compared based on fragment per kilobase transcript per million mapped reads (FPKM) values, and the star marks indicate significant differences (Tukey’s multiple comparison test, *p* < 0.01). (**B**) The phylogenetic analysis of genes was performed using MEGA11, and the analysis was performed using the neighbor-joining method at the amino acid level.

**Table 1 insects-15-00743-t001:** Used primer information.

Purpose	Primers	Sequence
For the mutation survey of *Nl-EST1*	Nl-EST1_5UTR-F1	TGCCGAGCCGTAGTTGATGAT
	Nl-EST1_5UTR-F2	TCGAGCATCTATCCTGCCTCTT
	Nl-EST1_ORF-R1	GGCCATAGTTTCCAGCAAAGTC
	Nl-EST1_ORF-R2	GTCAGGGTCATCGAGGAAATCT
	Nl-EST1_ORF-R3	GCTCCTGGGAAGTTCTTCTTCA
	Nl-EST1_3UTR-R1	GCCTACCTACCGTACTCAATTTTAATG
For the mutation survey of *ACE1*, G119A	Nl-ace1_G119A-F	CATGACTCGCACATCCTCAACA
	Nl-ace1_G119A-R	CTGCATGCTGACAAGTATGACG
For the mutation survey of *ACE1*, F331H	Nl-ace1_F331H-F	GGTCGTTGGCGACGAAAAACTT
	Nl-ace1_F331H-R	TGTAGAAACTCGTCCCGGTTGA

**Table 2 insects-15-00743-t002:** Bioassay results for three strains of *N. lugens* against a carbamate insecticide, fenobucarb.

Strains	LC_50_ Values (mg/L) (95% CI)	χ^2^ Log10 (Dose)	Resistance Ratio
BPH80	3.08 (2.69–3.51)	228.72	1
BPH15	10.06 (9.08–12.36)	551.7222	3.27
BPH19	73.98 (65.69–83.01)	441.83	24.02

Resistance ratio (RR) = LD50 of BPH15 or BPH19/LD50 of the susceptible strain, BP80.

**Table 3 insects-15-00743-t003:** Statistics of unigene contig.

Assembly	No. of Genes	No. of Transcripts	GC (%)	N50	Avg. Contig Length (bp)	Total Assembled Bases (bp)
merge	119,664	148,234	41	896	611	73,153,339
BPH80	75,069	92,362	40	816	579	43,504,047
BPH15	69,319	79,691	38	765	560	38,844,156
BPH19	93,427	111,118	41	860	600	56,127,492

Total trinity “genes”: total number of assembled genes by Trinity; total trinity transcripts: total number of assembled transcripts by Trinity.

**Table 4 insects-15-00743-t004:** Statistics of ORF prediction.

Assembly	Total Unigene	ORF Predicted Unigene	Single ORF Predicted Unigene	Multiple ORF Predicted Unigene
merge	119,664	30,788 (25.73%)	30,064 (97.65%)	724 (2.35%)
BPH80	75,069	18,593 (24.77%)	18,349 (98.69%)	244 (1.31%)
BPH15	69,319	15,880 (22.91%)	15,721 (99.0%)	159 (1.0%)
BPH19	93,427	26,147 (27.99%)	25,721 (98.37%)	426 (1.63%)

## Data Availability

The datasets generated during the current study are available from the corresponding author upon reasonable request.

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
