# Peer review of "RNAseq-Based Carboxylesterase Nl-EST1 Gene Expression Plasticity Identification and Its Potential Involvement in Fenobucarb Resistance in the Brown Planthopper Nilaparvata lugens"

_insects, 2024, doi:10.3390/insects15100743_

Round 1
Reviewer 1 Report
Comments and Suggestions for Authors
Reviewers' comments:
With the long-term and large-scale application of insecticides, BPH has evolved different levels of resistance to most insecticides such as imidacloprid, thiamethoxam, clothianidin, buprofezin, ethiprole and pymetrozine. Better understanding of resistance mechanisms in BPH can provide effective guidance for the management of this pest. This paper evaluates that a carboxylesterase (EST1) may be involved in fenobucarb resistance in the BPH. The authors showed that (1) Compare to the Sus strain, two field populations of BPH had developed different resistance levels to fenobucarb; (2) No mutation was found on the target gene Ace1 through sequencing; (3) Through RNA-Seq, it was found that there is a certain linear relationship between the expression level of EST1 and the resistance level of brown planthopper to fenobucarb. However, shortcomings in the experimental design and a lack of information in the experimental procedures are detrimental to the quality of the manuscript. Meanwhile, the experimental results have not been fully discussed.
The major comments were showed below:
1. The title is not appropriate and there are some issues. (1) The grammar of the title is incorrect; (2) This study only showed that two field populations of BPH have resistance to fenobucarb, and cannot represent all carbamate insecticides. (3) This is the most important point, as the whole article describes the situation of transcriptome sequencing from 3.3 to 3.8. Only in the last 3.8, it was found that the EST1 gene was highly expressed in the field population, and there was no further functional verification.
2. In the section of Bioassay. It is recommended to add a table to display important information such as slope, SE, 95% confidence interval, degrees of freedom, P-value, etc. regarding the bioassay results described in section 3.1.
3. In the mutation detection of the ace1 and Nl-EST1 genes, (1) the author only tested the known resistance sites G119A and F331H. What about the other regions of ace1 gene? If other regions of ace1 and ace2 cannot be excluded, the sentence cannot be described in the results “the fenobucarb resistance in these populations is not due to point mutations in the ace1 gene”. (2) It is recommended to add a table displaying the results of this section, including information such as the number of individuals tested, genotype, mutation frequency, etc. (3) The method description in this section is somewhat simple. It should be supplemented with information such as the number of samples detected for each population, how many BHPs each sample contains, etc. Usually, when conducting mutation detection, sequencing detection is performed on an individual, and at least 20 individuals are tested for each population.
4. In the section of de novo assembly of unigene sets. In Table S3, we can see that the gene number of initial assembled contig is smaller than the number of transcripts, but in Table 2, why is it that the gene number of unigene contig is actually the same as the number of transcripts? In addition, there seems to be a difference in the number of unigenes among these three brown planthopper populations, with BHP19 having the highest number of genes. This result can be explained and speculated in the discussion section.
5. In the section of the transcriptomic DEG analysis, the manuscript states that “These RNA-seq analyses suggest that the upregulation of genes present in BPH15 and BPH19 populations conferred fenobucarb resistance compared to BPH80.” The conclusion is too arbitrary. The results of the article only showed that BHP15 and BHP19 have high resistance to fenobucarb, and some differential genes were found.
6. In the section of the DEG analysis of detoxification genes. (1) “A strong correlation was observed between resistance levels and the expression of CCEs”, From the Figure 4B, CCEs may include XP_039289438, XP_039277448, XP_039277392 and XP_039280927. So why can only the XP_039277448 (EST1) gene be seen to have a linear relationship among the three populations in Figure 5? Are the FPKM values of genes used in transcriptome sequencing in Figures 4 and 5 different? (2) From figure 4, the genes of CYPs (XP_022200449, AIW79980 and AIW79957), UGT (XP_022189984), ABCs (XP_022195861, XP_039280719) and CPs (AWK28312, AWK28313 and AWK28311) may also have correlation between the expression and resistance levels, these genes also need to be validated through qPCR. (3) The number of genes compared by transcriptome is not complete, for example, there are more than 54 P450 genes in BPH, but only34 P450 genes were compared here, not all of them. So, it cannot be said “no correlation was found between resistance levels and the expression of CYPs, GSTs, UGTs, and ABCs in all strain and populations”.
7. In discussion, the description in this section is not very good. There is still too much elaboration on the results without providing more valuable perspectives on the results. Suggest rewriting this section. The following viewpoints can be supplemented with additional arguments, such as the existing resistance mechanisms of BPH to carbamate insecticides, the relationship between Ace mutations mediated resistance of insects or BPH to insecticides, and the resistance mechanisms of insects or BPH to insecticides involving carboxylesterase EST genes.
Minor comments:
1. Line 28 “test results” change to “bioassay results”.
2. Line 28-29 The unit of LC50 is not labeled, is it mg/L? Line211 has the same issue.
3. Line 32 “target site” change to “target gene” or “target of action”.
4. Line 55 The harm of BPH to rice also includes egg laying.
5. Line 69 “(glutathione-S-transferase, GST), which convert xenobiotics into non-toxic compounds” is not very accurate, endogenous molecules such as glutathione are conjugated with the first phase metabolites by glutathione-S-transferases (GSTs), further increasing the polarity and hydrophilicity of the products.
6. Line 100 “C. lectularius” > italic. Line 146 “N. lugens” > italic. And line 384.
7. Line 127 During the adult stage, male and female insects can already be clearly distinguished. How many males or females are there among the 10 adults? It should be written clearly here.
8. Line 138 How many males or females are there among the 20 adults? The expression levels of some genes are related to wing type and gender, so this information should be described clearly.
9. Line 152 “ace1 and Nl-EST1” > gene should be italicized. Line 220 etc.
10. Line 213 “Compared to BPH15, and BPH19 based on BPH80, exhibited…” change to “Compared to BPH80, BPH15 and BPH19 exhibited…”.
11. Line 285 “Table S4” change to “Table 2”, The data displayed here are all from Table 2.
12. Line 330 “hierarchical” and “log2 based” > typo
13. Line367 The clarity of Figure 3 is insufficient and a bit blurry.
14. Line 529, line 532, line 534 etc. The Latin names of insects in the references are not italicized, such as the “Nilaparvata lugens” and “Spodoptera exigua”.
15. Line 545 “Pest management science”, line 559 “Pest Management Science” etc. The capitalization of the journal's initial letters in the references is not consistent. Line 583 “Annu. Rev. Entomol.” Please change them according to the requirements of the Insects journal.
Author Response
Reviewers' comments:
With the long-term and large-scale application of insecticides, BPH has evolved different levels of resistance to most insecticides such as imidacloprid, thiamethoxam, clothianidin, buprofezin, ethiprole and pymetrozine. Better understanding of resistance mechanisms in BPH can provide effective guidance for the management of this pest. This paper evaluates that a carboxylesterase (EST1) may be involved in fenobucarb resistance in the BPH. The authors showed that (1) Compare to the Sus strain, two field populations of BPH had developed different resistance levels to fenobucarb; (2) No mutation was found on the target gene Ace1 through sequencing; (3) Through RNA-Seq, it was found that there is a certain linear relationship between the expression level of EST1 and the resistance level of brown planthopper to fenobucarb. However, shortcomings in the experimental design and a lack of information in the experimental procedures are detrimental to the quality of the manuscript. Meanwhile, the experimental results have not been fully discussed.
Responses:
Thank you for your thorough and constructive review of our manuscript. We appreciate your insightful feedback, which has greatly contributed to improving the quality of our work. In response to your comments, we have made several revisions to the manuscript using track changes to address the issues raised.
The major comments were showed below:
- The title is not appropriate and there are some issues. (1) The grammar of the title is incorrect; (2) This study only showed that two field populations of BPH have resistance to fenobucarb, and cannot represent all carbamate insecticides. (3) This is the most important point, as the whole article describes the situation of transcriptome sequencing from 3.3 to 3.8. Only in the last 3.8, it was found that the EST1gene was highly expressed in the field population, and there was no further functional verification.
Response 1: Thank you for your valuable feedback on the title of our manuscript. We agree that the title should be more precise and should accurately reflect the scope and findings of the study. In response to your detailed points, we have revised the title of the manuscript as follows:
RNAseq-based carboxylesterase Nl-EST1 gene expression plasticity identification and its potential involvement in fenobucarb resistance in the brown planthopper Nilaparvata lugens
- In the section of Bioassay. It is recommended to add a table to display important information such as slope, SE, 95% confidence interval, degrees of freedom, P-value, etc. regarding the bioassay results described in section 3.1.
Response 2: We appreciate your helpful suggestion to include a table presenting bioassay results in section 3.1. This table will assist readers in quickly comprehending the key metrics of the bioassay analysis. The table (Table 2) has been added to the revised manuscript on lines 233-236.
- In the mutation detection of the ace1and Nl-EST1 genes, (1) the author only tested the known resistance sites G119A and F331H. What about the other regions of ace1 gene? If other regions of ace1 and ace2 cannot be excluded, the sentence cannot be described in the results “the fenobucarb resistance in these populations is not due to point mutations in the ace1 gene”. (2) It is recommended to add a table displaying the results of this section, including information such as the number of individuals tested, genotype, mutation frequency, etc. (3) The method description in this section is somewhat simple. It should be supplemented with information such as the number of samples detected for each population, how many BHPs each sample contains, etc. Usually, when conducting mutation detection, sequencing detection is performed on an individual, and at least 20 individuals are tested for each population.
Response 3: We appreciate your feedback on the mutation detection in the ace1 and Nl-EST1 genes. Although our study focused on testing the known resistance sites G119A and F33H1, we conducted a comprehensive analysis of the entire open reading frame of the ace1 gene. Our methodology involved extracting pooled genomic DNA from 20 samples and conducting a thorough analysis rather than sequencing individual samples (Lines 168-170). This approach is supported by the paper by Kwon et al. 2022 (https://doi.org/10.1016/j.aspen.2012.08.001 and DOI: 10.1603/me09274). We also value your suggested approach and will consider using it in future research. In section 3.2, the mutation analysis (Line 237), Figure 1 contains all relevant data related to this section, so we kindly request that the table not be included here.
- In the section of de novo assembly of unigene sets. In Table S3, we can see that the gene number of initial assembled contig is smaller than the number of transcripts, but in Table 2, why is it that the gene number of unigene contig is actually the same as the number of transcripts? In addition, there seems to be a difference in the number of unigenes among these three brown planthopper populations, with BHP19 having the highest number of genes. This result can be explained and speculated in the discussion section.
Response 4: The number of unigenes differs between strains because the genetic background and the number of expressed genes differ between each strain, and this can be seen in other insect species as well. In our previous research group's study on the pyrethroid resistance mechanism of Helicoverpa armigera, there were cases where the number of unigenes differed depending on the strain, and I think this is a very natural result. For further confirmation please you can see the paper by Kim et al., 2024, Genome‑wide exploration of metabolic‑based pyrethroid resistance mechanism in Helicoverpa armigera
- In the section of the transcriptomic DEG analysis, the manuscript states that “These RNA-seq analyses suggest that the upregulation of genes present in BPH15 and BPH19 populations conferred fenobucarb resistance compared to BPH80.” The conclusion is too arbitrary. The results of the article only showed that BHP15 and BHP19 have high resistance to fenobucarb, and some differential genes were found.
Response 5: We are thankful for your meticulous point and agree that the conclusion presented in the manuscript could be further refined. We have revised the manuscript by adding the following clarification on lines 344-349.
Our current study evaluated differential gene expression in BPH80, BPH15, and BPH19 strains, indicating that these genes could a key role in the resistance of BPH19 to fenobucarb. However, further research is required to investigate whether these genes play a significant role in fenobucarb resistance. Our findings just suggest an association between gene expression patterns and resistance but do not establish causation.
- In the section of the DEG analysis of detoxification genes. (1) “A strong correlation was observed between resistance levels and the expression of CCEs”, From the Figure 4B, CCEs may include XP_039289438, XP_039277448, XP_039277392 and XP_039280927. So why can only the XP_039277448 (EST1) gene be seen to have a linear relationship among the three populations in Figure 5? Are the FPKM values of genes used in transcriptome sequencing in Figures 4 and 5 different? (2) From figure 4, the genes of CYPs (XP_022200449, AIW79980 and AIW79957), UGT (XP_022189984), ABCs (XP_022195861, XP_039280719) and CPs (AWK28312, AWK28313 and AWK28311) may also have correlation between the expression and resistance levels, these genes also need to be validated through qPCR. (3) The number of genes compared by transcriptome is not complete, for example, there are more than 54 P450 genes in BPH, but only34 P450 genes were compared here, not all of them. So, it cannot be said “no correlation was found between resistance levels and the expression of CYPs, GSTs, UGTs, and ABCs in all strain and populations”.
Response 6: We acknowledge your observation regarding the correlation between resistance and CCE gene expression. (1) In Figure 4B, our primary objective was to evaluate the involvement of various detoxification gene families such as CYP, CCE, GST, UGT, ABC, and CP in the elevation of fenobucarb resistance in BPH19. Our results suggested that among these families, the CCE genes are likely involved in conferring resistance to fenobucarb (Figure 4). Following this, we sought to identify which specific gene (s) within the CCE family might play a key role in enhancing resistance of BPH19. From our analysis, we found that XP_039277448 (NI-EST1) exhibits a particularly strong correlation with increased resistance in BPH19 to fenobucarb (Figure 5), suggesting that this gene plays a central role in the resistance mechanism of BPH19 to fenobucarb. To clarify, the FPKM values used in the transcriptome sequencing in Figures 4 and 5 are the same. Figure 4 provides a broad comparison across detoxification gene families, while Figure 5 focuses on identifying the most relevant CCE gene contributing to resistance. (2) We agree with you that genes from other families, CYPs (XP_022200449, AIW79980, AIW79957), UGT (XP_022189984), ABCs (XP_022195861, XP_039280719), and CPs (AWK28312, AWK28313, AWK28311), could also potentially correlate with resistance levels, as suggested by Figure 4. While RNA-seq analysis of these genes, validating them through qPCR would be the next step to confirm their role in the resistance mechanisms. However, as stated in your response, the focus of the current study is to identify key genes based on RNA-seq data, with particular emphasis on the CCE family and its involvement in fenobucarb resistance. Future work can focus on validating the expression of the identified genes through qPCR. (3) It is noted that the transcriptomic analysis does not cover the entire detoxification gene family, particularly the CYP P450 family, where only 34 out of 54 genes were compared. This limits the ability to make broad conclusions about the entire family. Therefore, the claim “no correlation was found between resistance levels and the expression of CYPs, GSTs, UGTs, and ABCs in all BPH strains and populations” has been revised in the manuscript as “no correlation was found between the resistance levels of BPH19 and the expression of 34 out of the 54 CYP genes, as well as all GST, UGT, and ABC genes analyzed.” See lines 413-415.
- In discussion, the description in this section is not very good. There is still too much elaboration on the results without providing more valuable perspectives on the results. Suggest rewriting this section. The following viewpoints can be supplemented with additional arguments, such as the existing resistance mechanisms of BPH to carbamate insecticides, the relationship between Acemutations mediated resistance of insects or BPH to insecticides, and the resistance mechanisms of insects or BPH to insecticides involving carboxylesterase EST genes.
Response 7: We are thankful for your valuable suggestions, and based on them, we have carefully revised the Discussion section of the manuscript.
Minor comments:
- Line 28 “test results” change to “bioassay results”.
- Line 28-29 The unit of LC50is not labeled, is it mg/L? Line211 has the same issue.
Responses: We appreciate your thorough review of the manuscript, and we have made the necessary corrections based on your suggestions.
- Line 32 “target site” change to “target gene” or “target of action”.
Response 3: While the term “target site” is commonly used in toxicology to refer proteins, genes, etc., that xenobiotics interact with, we understand that clarity and consistency in the manuscript are important. To avoid any confusion, we have replaced “target site” with “target gene” in the sentence, as it more precisely refers to the acetylcholinesterase 1 gene in this context. Thank you.
- Line 55 The harm of BPH to rice also includes egg laying.
Response 4: We also included the “egg laying” in the revised manuscript. Thank you.
- Line 69 “(glutathione-S-transferase, GST), which convert xenobiotics into non-toxic compounds” is not very accurate, endogenous molecules such as glutathione are conjugated with the first phase metabolites by glutathione-S-transferases (GSTs), further increasing the polarity and hydrophilicity of the products.
Response 5: Thank you for pointing this point. We have revised the statement to be more accurate. The updated sentences are “In the first phase of metabolism, monoxygenases and hydrolases modify xenobiotics, making them more reactive. In the second phase, GSTs conjugate endogenous molecules such as glutathione to these metabolites, increasing their polarity and hydrophilicity. This process facilitates their excretion and aids in detoxification”. We hope that this revision provides a clear and more precise description.
- Line 100 “C. lectularius” > italic. Line 146 “N. lugens” > italic. And line 384.
Responses 6: We appreciate your thorough review of the manuscript, and we have made the necessary corrections based on your suggestions.
- Line 127 During the adult stage, male and female insects can already be clearly distinguished. How many males or females are there among the 10 adults? It should be written clearly here.
Response 7: Thank you for your comment. We have clarified this in the revised manuscript, specifying that all 10 adults were wingless females.
- Line 138 How many males or females are there among the 20 adults? The expression levels of some genes are related to wing type and gender, so this information should be described clearly.
Response 8: Thank you for your comment. We have clarified this in the revised manuscript, specifying that all 20 adults were wingless females.
- Line 152 “ace1 and Nl-EST1” > gene should be italicized. Line 220 etc.
- Line 213 “Compared to BPH15, and BPH19 based on BPH80, exhibited…” change to “Compared to BPH80, BPH15 and BPH19 exhibited…”.
- Line 285 “Table S4” change to “Table 2”, The data displayed here are all from Table 2.
- Line 330 “hierarchical” and “log2 based” > typo
Responses: We appreciate your thorough review of the manuscript, and we have made the necessary corrections based on your suggestions.
- Line367 The clarity of Figure 3 is insufficient and a bit blurry.
- Line 529, line 532, line 534 etc. The Latin names of insects in the references are not italicized, such as the “Nilaparvata lugens” and “Spodoptera exigua”.
Responses: We appreciate your thorough review of the manuscript, and we have made the necessary corrections throughout the manuscript based on your suggestions.
- Line 545 “Pest management science”, line 559 “Pest Management Science” etc. The capitalization of the journal's initial letters in the references is not consistent. Line 583 “Annu. Rev. Entomol.” Please change them according to the requirements of the Insects journal.
Response 15: We appreciate your thorough review of the manuscript, and we have made the necessary corrections based on your suggestions throughout the manuscript.

Reviewer 2 Report
Comments and Suggestions for Authors
I have two suggestions:
1) Please request the authors to demonstrate molecularly that BPH80, BPH15 and BPH19 are specimens belonging to the same phylogenetic insect species
2) In the discussion include the concepts of epigenesis, species complex, perhaps sympatric biological species in the context of the present data and with the demonstration that the BPH specimens belong to the same phylogenetic species

No comments
Author Response
Reviewer 2
We sincerely appreciate your thorough and constructive review of our manuscript. Your valuable suggestions have significantly contributed to enhancing the quality and clarity of our work. In response to your insightful comments, we have revised the manuscript and highlighted all changes using track changes for your convenience. We believe these revisions address your concerns and improve the overall manuscript.
Thank you once again for your time and thoughtful feedback.
Comments
- Please request the authors to demonstrate molecularly that BPH80, BPH15 and BPH19 are specimens belonging to the same phylogenetic insect species
Response 1: The BPH populations (in the revised manuscript we have replaced the term populations with the term strains to avoid confusion) used in this study were identified and confirmed as belonging to the same species using the mitochondrial cytochrome oxidase 1 (mtCO1) gene marker, a widely accepted molecular tool for species identification. This method provides a high degree of accuracy in distinguishing between closely related species. Additionally, Since the field populations were not collected directly for this study, the risk of species mixing or contamination is minimal. Therefore, we are confident that BPH80, BPH15, and BPH19 all belong to the same phylogenetic species, reducing the possibility of analysis errors due to species misidentification. Please see lines 125-130.
- In the discussion include the concepts of epigenesis, species complex, perhaps sympatric biological species in the context of the present data and with the demonstration that the BPH specimens belong to the same phylogenetic species
Response 2: Thank you for your valuable suggestions. We have carefully considered your advice and have revised the manuscript accordingly.

Reviewer 3 Report
Comments and Suggestions for Authors
The manuscript titled “Identification of the carboxylesterase, Nl-EST1 gene expression regulation possibly involved in carbamate insecticide resistance in brown planthopper Nilaparvata lugens” investigates the cause of higher resistance to fenobucarb in brown planthopper (Nilaparvata lugens) in the 2019 field population by comparing it with populations collected in the 1980 and 2015. The authors found no target site mutations in the acetylcholinesterase 1 gene. However, the authors identified an association between the level of expression of carboxylesterase 1, 22 NI-EST1 gene and carbamate insecticide resistance and suggested that Nl-EST1 could serve as an expression marker for resistance diagnosis.
The manuscript presents an important study on fenobucarb resistance in brown planthopper (BPH) Nilaparvata lugens by comparing susceptible and resistant populations from different years. While the overall approach and findings are interesting, several areas in the transcriptome data analysis need to be clarified before the paper can be recommended for publication.
Major points for revision:
1. Line 2.3 RNA and DNA extraction: It should describe when the samples were collected, if samples were exposed to fenobucarb, what was the dose of fenobucarb? how long was the exposure, and if there was any sample with no fenobucarb exposure as controls?
2. Line 157 RNA-seq analysis details: In Table 3, there are significant differences in the total Unigene, single ORF predicted Unigene among BPH80, BPH15 and BPH19. This is unexpected given the read depth per sample. I would suggest the authors evaluate sequence read contamination by a) aligning each the RNA-seq raw data to the existing genome assembly of ASM1435652v1; b) checking each Unigene has a hit of insect species; c) estimating how many Unigenes present only in one sample, and how many Unigenes present three samples within a group, but not in the other two groups. This process can provide a good assessment of the reference gene list for further analysis.
3. Line 309 3.6. Transcriptomic DEG analysis: The key output of Transcriptome DEG analysis is to find differentially expressed transcripts between samples. The authors listed the clustering, PCA and MA plot, but missed the main table of the output.
4. Line 339 “Following exposure to fenobucarb, we noted significant disparities in cluster counts.” This was not described in the Methods and Materials.
5. Line 340-342 “The BPH19 population, which 340 exhibited higher resistance, showed a notably higher cluster count (16,708) compared to 341 moderately resistant BPH15 (15,059) and the susceptible BPH80 (12,593), as illustrated in 342 Figure 3A.” The text description does not correspond to the figure, where BPH80 has a cluster count of 15,059.
6. Line 371 DEG analysis of detoxification genes: These results should come from the 3.6 Transcriptomic DEG analysis. The authors should explain why they need to do the DEG detoxification separately. They should provide a discussion point on the difference between these two methods.
7. Line 493 5. Conclusions. There are two ways to validate “the upregulation of E4 Type esterase gene NI-EST1 contributed to the elevated resistance of BPH19 in comparison to BPH80 and BPH15” 1) a quick RT-PCR to examine the expression of the candidate genes, 2) Synergist bioassays to confirm the metabolic resistance.
In conclusion, while the study presents potentially important findings regarding the role of Nl-EST1 in fenobucarb resistance in N. lugens, the bioinformatics data analysis methodology needs substantial revision and expansion. The validation of the results should not involve complex experiments. Addressing these points will greatly strengthen the manuscript and increase its impact in the field of pesticide resistance research.
Comments on the Quality of English LanguageThe language of the manuscript is acceptable.
Author Response
Reviewer 3
The manuscript titled “Identification of the carboxylesterase, Nl-EST1 gene expression regulation possibly involved in carbamate insecticide resistance in brown planthopper Nilaparvata lugens” investigates the cause of higher resistance to fenobucarb in brown planthopper (Nilaparvata lugens) in the 2019 field population by comparing it with populations collected in the 1980 and 2015. The authors found no target site mutations in the acetylcholinesterase 1 gene. However, the authors identified an association between the level of expression of carboxylesterase 1, 22 NI-EST1 gene and carbamate insecticide resistance and suggested that Nl-EST1 could serve as an expression marker for resistance diagnosis.
The manuscript presents an important study on fenobucarb resistance in brown planthopper (BPH) Nilaparvata lugens by comparing susceptible and resistant populations from different years. While the overall approach and findings are interesting, several areas in the transcriptome data analysis need to be clarified before the paper can be recommended for publication.
Response: We sincerely thank you for your thorough review of our manuscript and for your positive feedback on our work. In response to your valuable comments, we have revised the manuscript accordingly, with all changes tracked for your convenience. We greatly appreciate the time and effort you have dedicated to reviewing our manuscript, and we are confident that your insightful suggestions have significantly enhanced the quality and clarity of our work.
Major points for revision:
- Line 2.3 RNA and DNA extraction: It should describe when the samples were collected, if samples were exposed to fenobucarb, what was the dose of fenobucarb? how long was the exposure, and if there was any sample with no fenobucarb exposure as controls?
Response 1: We appreciate your meticulous attention to detail. In response to your suggestion, we have included information regarding the timing of sample collection, fenobucarb exposure, dosage, duration of exposure, and the use of non-exposed control samples in the revised manuscript. Please see the updates on lines 124–129.
- Line 157 RNA-seq analysis details: In Table 3, there are significant differences in the total Unigene, single ORF predicted Unigene among BPH80, BPH15 and BPH19. This is unexpected given the read depth per sample. I would suggest the authors evaluate sequence read contamination by a) aligning each the RNA-seq raw data to the existing genome assembly of ASM1435652v1; b) checking each Unigene has a hit of insect species; c) estimating how many Unigenes present only in one sample, and how many Unigenes present three samples within a group, but not in the other two groups. This process can provide a good assessment of the reference gene list for further analysis.
Response 2: We appreciate your insightful observation regarding the unexpected differences in total Unigene and single ORF predicted Unigene counts among BPH80, BPH15, and BPH19.
In our previous paper on the pyrethroid insecticide resistance mechanism of Helicoverpa armigera, we performed two DEGs.
Kim et al., 2024, Genome‑wide exploration of metabolic‑based pyrethroid resistance mechanism in Helicoverpa armigera, Journal of Pest Science https://doi.org/10.1007/s10340-024-01797-8
First, reference genome-based DEGs were used, and second, unigene set-based DEGs were used. Of course, the method you suggested is quite ideal. However, in the case of DEG based on the reference genome, some genes may be excluded from the analysis depending on the reference genome quality or the annotation completeness. Therefore, in this study, we focused on comparing the differences between three BPHs with different genetic backgrounds, so we created a unigene set through de novo assembly and performed DEG analysis based on this. In addition, to confirm the differences by strain using orthovenn3, we created a unigene set for each strain and compared them.
- Line 309 3.6. Transcriptomic DEG analysis: The key output of Transcriptome DEG analysis is to find differentially expressed transcripts between samples. The authors listed the clustering, PCA and MA plot, but missed the main table of the output.
Response 3: Thank you for pointing out the omission in the presentation of the key output from the transcriptomic DEG analysis. We agree that the main table of differentially expressed genes (DEGs) is essential for understanding the results.
However, organizing all differentially expressed genes into a table would be extensive data depending on how we set the criteria. So, we organized it into a specific section centered on detoxification enzyme genes.
- Line 339 “Following exposure to fenobucarb, we noted significant disparities in cluster counts.” This was not described in the Methods and Materials.
Response 4: That’s our mistake. We rewrite as following sentence.
We noted significant disparities in cluster counts according to the level of fenobucarb resistance on lines 363 and 364.
- Line 340-342 “The BPH19 population, which 340 exhibited higher resistance, showed a notably higher cluster count (16,708) compared to 341 moderately resistant BPH15 (15,059) and the susceptible BPH80 (12,593), as illustrated in 342 Figure 3A.” The text description does not correspond to the figure, where BPH80 has a cluster count of 15,059.
Response 5:
Thank you for bringing this discrepancy to our attention. We have reviewed the figure and identified the inconsistency between the text and the data presented. The cluster counts for BPH80, BPH15, and BPH19 were incorrectly described in the text. We have corrected the text to accurately reflect the data shown in Figure 3A. In the revised manuscript, the cluster count for BPH80 has been updated to 15,059, in line with the figure on line 368 of the revised manuscript. We apologize for the oversight and appreciate your attention to detail.
- Line 371 DEG analysis of detoxification genes: These results should come from the 3.6 Transcriptomic DEG analysis. The authors should explain why they need to do the DEG detoxification separately. They should provide a discussion point on the difference between these two methods.
Response 6: Thank you for your observation. To clarify, section 3.6 (Transcriptomic DEG analysis) (lines 332–339) provides a comprehensive overview of the total gene expression patterns across BPH80, BPH15, and BPH19, highlighting the overall differences in gene expression among the populations. This broader analysis is intended to provide a general context for the transcriptomic changes observed. In contrast, section 3.6 (DEG analysis of detoxification genes) (lines 370–377) focuses specifically on detoxification genes. This separate analysis is aimed at examining the expression patterns of genes involved in detoxification processes to identify which specific gene systems are most closely associated with resistance induction in BPH. By isolating the detoxification genes, we can better understand their role and significance in resistance mechanisms. We have added the reasons why we conducted these analyses separately on lines 408-415.
- Line 493 5. Conclusions. There are two ways to validate “the upregulation of E4 Type esterase gene NI-EST1 contributed to the elevated resistance of BPH19 in comparison to BPH80 and BPH15” 1) a quick RT-PCR to examine the expression of the candidate genes, 2) Synergist bioassays to confirm the metabolic resistance.
Response 7: While our study primarily addresses transcriptomic changes, we have added a statement in the conclusion section (lines 594–599) acknowledging the need for such further validation. The revised conclusion now reads: “Though our current investigation provides valuable insights into transcriptomic changes associated with fenobucarb resistance in N. lugens in Korea, further validation through RT-PCR and synergist bioassays is needed to confirm the role of the E4 type esterase gene Nl-EST1 in contributing to the elevated resistance observed in the BPH19 strain.”
In conclusion, while the study presents potentially important findings regarding the role of Nl-EST1 in fenobucarb resistance in N. lugens, the bioinformatics data analysis methodology needs substantial revision and expansion. The validation of the results should not involve complex experiments. Addressing these points will greatly strengthen the manuscript and increase its impact in the field of pesticide resistance research.
Response: Thank you for your valuable feedback. We agree that the bioinformatics data analysis methodology is crucial for the robustness of our findings. We will address the suggested revisions and expand on the methodology to ensure a more comprehensive and rigorous analysis. Additionally, we acknowledge the importance of validating results practically and will ensure that our validation approaches are straightforward and directly relevant to the findings. We believe that these revisions will indeed strengthen the manuscript and enhance its contribution to the field of pesticide resistance research.

Round 2
Reviewer 1 Report
Comments and Suggestions for Authors
I have cautiously revised the authors’ responses and rebuttals to my comments and I am satisfied with their feedback. The manuscript is now more precise on the title and discussion and clearer. The conclusions made are now supported by the data provided. However, there is still one question that has not been full answered, and I comment below.
Question 4: In the section of de novo assembly of unigene sets. In Table S3, we can see that the gene number of initial assembled contig is smaller than the number of transcripts, but in Table 2 (The revised version has been updated to Table 3), why is it that the gene number of unigene contig is actually the same as the number of transcripts? In addition, there seems to be a difference in the number of unigenes among these three brown planthopper populations, with BHP19 having the highest number of genes. This result can be explained and speculated in the discussion section.
Authors’ response: The number of unigenes differs between strains because the genetic background and the number of expressed genes differ between each strain, and this can be seen in other insect species as well. In our previous research group's study on the pyrethroid resistance mechanism of Helicoverpa armigera, there were cases where the number of unigenes differed depending on the strain, and I think this is a very natural result. For further confirmation please you can see the paper by Kim et al., 2024, Genome‑wide exploration of metabolic‑based pyrethroid resistance mechanism in Helicoverpa armigera.
Reviewer’s response: The author only explains the second half of the sentence here, while in Table 2 (The revised version has been updated to Table 3), the number of genes and transcripts in each population is the same, which does not explain why. In theory, a gene may correspond to multiple transcripts, so the number of genes is less than the number of transcripts.
Minor comments:
1. As this version is still a revised manuscript, not a clean version. For example, in lines 452-542 of the discussion, some complete sentences cannot be clearly seen. So please carefully revise any possible details before the article is accepted. For example, Line 473, “BPH1980” change to “BPH80”.
2. Line 556, “that metabolic”, there seems to be an extra space between these two words.
Author Response
I have cautiously revised the authors’ responses and rebuttals to my comments and I am satisfied with their feedback. The manuscript is now more precise on the title and discussion and clearer. The conclusions made are now supported by the data provided. However, there is still one question that has not been full answered, and I comment below.
Question 4: In the section of de novo assembly of unigene sets. In Table S3, we can see that the gene number of initial assembled contig is smaller than the number of transcripts, but in Table 2 (The revised version has been updated to Table 3), why is it that the gene number of unigene contig is actually the same as the number of transcripts? In addition, there seems to be a difference in the number of unigenes among these three brown planthopper populations, with BHP19 having the highest number of genes. This result can be explained and speculated in the discussion section.
Authors’ response: The number of unigenes differs between strains because the genetic background and the number of expressed genes differ between each strain, and this can be seen in other insect species as well. In our previous research group's study on the pyrethroid resistance mechanism of Helicoverpa armigera, there were cases where the number of unigenes differed depending on the strain, and I think this is a very natural result. For further confirmation please you can see the paper by Kim et al., 2024, Genome‑wide exploration of metabolic‑based pyrethroid resistance mechanism in Helicoverpa armigera.
Reviewer’s response: The author only explains the second half of the sentence here, while in Table 2 (The revised version has been updated to Table 3), the number of genes and transcripts in each population is the same, which does not explain why. In theory, a gene may correspond to multiple transcripts, so the number of genes is less than the number of transcripts.
Response: We appreciate you for highlighting this critical point. We agree that, in theory, the number of transcripts should always exceed the number of genes, as a single gene can give rise to multiple transcripts. As can be confirmed in Table S3, Statistics of initially assembled contigs, genes were selected from the entire transcripts and reorganized based on this as in Table 3, Statistics of unigene contigs. Please refer to lines 302 and 303 of the updated manuscript for the relevant changes.
Minor comments:
- As this version is still a revised manuscript, not a clean version. For example, in lines 452-542 of the discussion, some complete sentences cannot be clearly seen. So please carefully revise any possible details before the article is accepted. For example, Line 473, “BPH1980” change to “BPH80”.
- Line 556, “that metabolic”, there seems to be an extra space between these two words.
Response: We sincerely appreciate the thorough review of our manuscript. We have carefully reviewed the entire manuscript, particularly the discussion section, to eliminate any confusion. Additionally, we have corrected the pointed words and removed any unnecessary spaces in the text.
Reviewer 2 Report
Comments and Suggestions for Authors
insects-3179277 review 2nd round
Authors are requested to address the following issues:
1) On this occasion, the title of the manuscript changed and included the word "plasticity", I guess it's "genic" This changed the whole meaning of the intention to communicate the purpose of the authors. Therefore, the authors are requested to establish the congruence between the title, the simple summary, abstract, in last paragraph of the introduction section and categorically in the conclusion.
2) Please include the database and accession number of these insect species (gene). Preferably, the database should be GenBank from NIH-USA.

No comments
Author Response
- On this occasion, the title of the manuscript changed and included the word "plasticity", I guess it's "genic" This changed the whole meaning of the intention to communicate the purpose of the authors. Therefore, the authors are requested to establish the congruence between the title, the simple summary, abstract, in last paragraph of the introduction section and categorically in the conclusion.
Response: Thank you for highlighting critical issue. We have revised the manuscript title based on the suggestion of one of the reviewers, and the revised title has been agreed upon (please refer to our original manuscript title for comparison). As a result, we have also adjusted the simple summary abstract, the last paragraph of the introduction, and the conclusion to ensure consistency throughout the manuscript.
Please include the database and accession number of these insect species (gene). Preferably, the database should be GenBank from NIH-USA.
Response: Thank you for the suggestion. We have included database and accession numbers for the insect species (genes) in the revised manuscript, with a preference for Gen Bank from NIH-USA.
Reviewer 3 Report
Comments and Suggestions for Authors
Reviewer’s Report for revised manuscript.
The author has addressed most of comments to the initial reviewer’s report and made substantial improvements to the manuscript. However, there are still some important issues that need to be addressed before the manuscript can be recommended for publication.
Responded to comment 2:
b) checking each Unigene has a hit of insect species;
c) estimating how many Unigenes present only in one sample, and how many Unigenes present three samples within a group, but not in the other two groups. This process can provide a good assessment of the reference gene list for further analysis.
The authors’ response to comment 3 is not entirely satisfactory. While it’s understandable that presenting all DEGs in a table might be extensive, the following points need to be addressed:
a) The manuscript should clearly state the criteria used to classify genes as differentially expressed in the DEG analysis.
b) The authors should provide a comprehensive DEG table as a supplementary file.
c) The manuscript should compare the trends observed in the global DEG analysis with those found in the detoxification gene-specific analysis. This comparison would strengthen the findings and provide a more comprehensive view of the transcriptomic changes.
Once these revisions are made, the manuscript should be suitable for publication. The authors’ efforts in improving the manuscript are appreciated, and these final revisions will ensure that the study makes a valuable contribution to the field.
Author Response
The author has addressed most of comments to the initial reviewer’s report and made substantial improvements to the manuscript. However, there are still some important issues that need to be addressed before the manuscript can be recommended for publication.
Responded to comment 2:
- b) checking each Unigene has a hit of insect species;
- c) estimating how many Unigenes present only in one sample, and how many Unigenes present three samples within a group, but not in the other two groups. This process can provide a good assessment of the reference gene list for further analysis.
The authors’ response to comment 3 is not entirely satisfactory. While it’s understandable that presenting all DEGs in a table might be extensive, the following points need to be addressed:
- a) The manuscript should clearly state the criteria used to classify genes as differentially expressed in the DEG analysis.
- b) The authors should provide a comprehensive DEG table as a supplementary file.
Response: Thank you for the suggestion. We have now included a comprehensive DEG table as a supplementary Excel file, providing detailed information on all differentially expressed genes identified in the study. This addition allows for a more in-depth examination of the gene expression data and enhances the transparency and reproducibility of our findings.
- c) The manuscript should compare the trends observed in the global DEG analysis with those found in the detoxification gene-specific analysis. This comparison would strengthen the findings and provide a more comprehensive view of the transcriptomic changes.
Response: Thank you for the valuable feedback. We have revised the manuscript to compare better the trends observed in the global DEG analysis with the detoxification gene-specific analysis, as suggested. We have added it in the revised manuscript on lines 489-500. The added lines are “The results from Figures 2, 4, and 5 highlight the importance of using global and targeted gene expression analysis to understand resistance mechanisms better. In Figure 2, the global analysis of DEG showed distinct expression patterns among all BPH strains. BPH19 exhibited significant differences in gene expression compared to BPH80 and BPH15, suggesting its potential involvement in fenobucarb resistance. However, this broad analysis did not identify specific resistance-related genes or pathways. On the other hand, the focused detoxification gene analysis (Figures 4 and 5) showed a strong correlation between the upregulation of the E4-type CCE gene Nl-EST1 and fenobucarb resistance, particularly in BPH19, followed by BPH15. This comparison demonstrates how combining global transcriptomic and targeted analysis can provide a comprehensive overview of gene expression and a more precise identification of key resistance-related genes, such as Nl-EST1, in BPH strains.”
Once these revisions are made, the manuscript should be suitable for publication. The authors’ efforts in improving the manuscript are appreciated, and these final revisions will ensure that the study makes a valuable contribution to the field.
Response: Thank you for the warm comments. We tried to edit this manuscript so that it is suitable for publication. Thank you very much once again.